# Time Pressure Weakens Social Norm Maintenance in Third-Party Punishment

**DOI:** 10.3390/brainsci13020227

**Published:** 2023-01-29

**Authors:** Xing Zhou, Yanqing Wang, Weiqi He, Shuaixia Li, Shuxin Jia, Chunliang Feng, Ruolei Gu, Wenbo Luo

**Affiliations:** 1Research Center of Brain and Cognitive Neuroscience, Liaoning Normal University, Dalian 116029, China; 2Key Laboratory of Brain and Cognitive Neuroscience, Liaoning Province, Dalian 116029, China; 3Institute of Psychology, Chinese Academy of Sciences, Beijing 100101, China; 4School of Psychology, South China Normal University, Guangzhou 510631, China

**Keywords:** time pressure, third-party punishment, pro-social, pro-self, social norms

## Abstract

Decision-making under time pressure may better reflect an individual’s response preference, but few studies have examined whether individuals choose to be more selfish or altruistic in a scenario where third-party punishment is essential for maintaining social norms. This study used a third-party punishment paradigm to investigate how time pressure impacts on individuals’ maintenance of behavior that follows social norms. Thirty-one participants observed a Dictator Game and had to decide whether to punish someone who made what was categorized as a high unfair offer by spending their own Monetary units to reduce that person’s payoff. The experiment was conducted across different offer conditions. The study results demonstrated that reaction times were faster under time pressure compared with no time pressure. Time pressure was also correlated with less severe punishment. Specifically, participants were less likely to punish the dictator under time pressure compared with no time pressure when the offer was categorized as a high unfair. The findings suggested that individuals in these game conditions and under time pressure do not overcome their pro-selves and that time pressure weakens an individual’s willingness to punish high unfair offers.

## 1. Introduction

When others violate the principle of fairness, research has shown that third-party punishment with the aim of maintaining social norms is a more common means of guaranteeing social justice than coercive legal means [1] and this punishment is also a representative altruistic behavior [2,3]. In this game-based scenario, the punishment is imposed on the norm violator by a third party who is unrelated to both interested parties. Individuals are willing to incur costs to punish the norm violator even when the norm violation does not directly harm the third party [1,4]. This punishment process is also known as costly punishment or altruistic punishment. Compared with second-party punishment, in third-party punishment the third party is not a direct beneficiary and can punish norm violators fairly and reasonably based on an objective fact. Additionally, third-party punishment requires incurring and paying for the costs to maintain social justice norms; thus, it is clearly altruistic in nature. Therefore, many studies have used third-party punishment to study the altruistic behavior of humans.

When faced with an unfair offer, a third party chooses to punish the proposer for the purpose of maintaining social norms. Even if the punishment is costly and the third-party does not gain any benefit directly or indirectly, they will still choose to punish the unfair proposer in order to maintain fairness [5]. The magnitude of third-party punishment increases as the violator’s fairness-norm violation worsens, owing to the third party’s inequity aversion [6,7,8,9]. Although social norm maintenance behavior in a third-party punishment context is influenced by factors such as motivation to maintain social norms and recipient status, previous studies have only explored external factors. Existing studies have not explored whether an individual’s intuitive choice is more pro-social or pro-ego when maintaining social norms.

There are costs to an individual’s own interests in maintaining justice through third-party punishment, which requires people to make decisions based on balancing pro-self and pro-social conflicts [10]. Decision finalization requires the integration of cognitive resources and multi-perspective information to make a rational decision within a limited time [11]. Pro-social behavior requires the integration of multiple perspectives about information in a limited time to make rational decisions; therefore, time is one of the important factors influencing the pro-social decision [10,11,12]. Researchers have introduced time pressure to explore pro-social behavior in human beings [12,13,14,15,16]. 

Time pressure refers to individuals’ subjective perception that they do not have enough time to complete tasks [17]. The conflict between selfishness and pro-sociality is heightened by time pressure that prevents individuals from integrating decision-relevant information systematically and weighing the pros and cons [18]. Although pro-social behavior requires people to balance the conflict between selfishness and pro-sociality to ensure self-interest and the interests of others, decision-making time constraints add another difficulty [15]. Individuals who experience time pressure and have insufficient time to overcome selfish tendencies have been shown to behave more selfishly and become less generous [15,19,20]. Dual-process theory assumes that intuitive choice is automatic and quick, whereas a deliberate decision is slow and requires more effort to integrate information [21]. Individuals under time pressure are unable to deliberate and effectively integrate decision-related information, which results in widely varying decisions. Therefore, the authors believe that time constraints can impede the process of deliberate reflection. Restrictions on the duration of participants’ objective choices can effectively explore people’s propensity to choose in social decision-making circumstances, which has important implications for whether individuals as third parties inherently uphold social norms.

As third-party punishment requires the loss of one’s own benefit to maintain social norms, it is the most representative altruistic behavior [22] and more directly reflects individual decision preferences under time pressure. Therefore, it is essential to explore the relationship between time pressure and third-party altruistic decision-making to both deepen the understanding of how socially normative behavior is maintained and to provide guidance on overcoming time constraints in making rational decisions in an informational society. This study examined the effect of time pressure on individuals’ behavior maintaining social norms by setting three levels (Fair, Medium unfair, and High unfair) within a game-based, third-party punishment paradigm. Based on inequity aversion theory, we hypothesized that when participants were involved in the dictator game task as a third party, punishment for the dictator would increase with levels of unfairness and that individuals under time pressure would punish the dictator more for an unfair offer than they would when under no time pressure.

## 2. Methods

### 2.1. Participants

Thirty-one healthy participants (15 female, mean age ± sd: 21.5 ± 3.7) were recruited from Liaoning Normal University. All participants had normal or corrected-to-normal vision and were right-handed. None of the participants had been approached with a similar experimental task or similar material prior to the experiment. We paid the participants CNY 25–35 after the experiment was completed. The sample size of this study was calculated using G*Power 3.1.9 [23]. According to the analysis (*d* = 0.25, *α* = 0.05, *β* = 0.95, analysis of variance (ANOVA), repeated measures, within factors), a total sample size of 28 participants was required to detect a reliable effect. This experiment was approved by the Ethics Committee of the Brain and Cognitive Neuroscience Research Center of Liaoning Normal University, and all participants gave informed consent prior to the experiment.

### 2.2. Experiment Design 

A 2 × 3 within-participant factorial design was used for the experiment, with the first factor being the time condition (Time pressure vs. no time pressure) and the second factor being the offer (Fair, 10:10; Medium unfair,12:8; and High unfair, 9:1). A total of six conditions were generated (Time pressure—Fair; Time pressure—Medium unfair; Time pressure—High unfair; No time pressure—Fair; No time pressure—Medium unfair; No time pressure—High unfair). For the time pressure, we separately recruited 14 participants to complete the third-party punishment task, which had the same procedure as the formal experimental task except for the stimulus duration. This time-pressure setting method is commonly used in time pressure studies [24,25]. There was no limitation on the duration of stimulation in any time pressure condition.

### 2.3. Procedure 

The behavioral data acquisition of the third-party punishment game was performed using E-Prime 2.0 software (Psychology Software Tools, Inc., Sharpsburg, PA, USA). To ensure the reliability of the task, groups of three participants (who did not know each other) began a third-party punishment task simultaneously in three different labs. While the participants were aware of the presence of strangers in the other two labs, they did not meet the others before or during the experiment. Before the experiment began, participants were informed that there were two other people performing the task. Those participants were selected randomly to be the dictator (Player A) or recipient (Player B). Player B had to accept Player A’s offer to complete a distribution task together. The task required the third participant (Player C, the third party) to decide whether to spend their Monetary units (MUs) to punish the dictator after observing the dictator’s offer to the recipient. Player C held four MUs per turn. Player C was informed that the final payment for all would be determined after the task was completed and would be based on their choice. In fact, all participants completed the task as third-party decision-makers (Player C).

At the start of each trial, a black fixation cross appeared at the center of the screen for 1000 ms. Player A’s offer was then displayed for Player C to see. At this stage, the participant had to decide whether to spend their MUs to punish the dictator. To choose to punish the dictator, Player C pressed 1 and to choose to retain their MUs, Player C pressed 2. It should be noted that when the screen displayed “Count Down” and the duration of the stimulus was refreshed from 1000 ms to 0 ms, there was a time pressure variable in the environment and participants had a time limit. When the screen displayed “Count Up” and counting began at 0 ms and continued until the participant made a choice, there was no time pressure and the participant did not have a time limit. If the participant chose to punish the dictator and indicated that onscreen, after 400–600 ms, the participant was directed to press 1–4 to indicate how many of their own MUs would be used to reduce the total amount of Player A’s money, which effectively punished Player A. Each MU spent by the participant reduced Player A’s total assets by three Mus. Finally, the number of Mus lost by the dictator (Player A) and the decision-maker (Player C) in the round was disclosed. If the participant chose to keep their MUs, Player A (the dictator) and Player C’s Mus for the round were displayed. If Player C did not make a choice in either condition, the statement “You didn’t make a choice” was displayed. Player C’s payment was calculated as a constant basic payment (CNY 10) plus the money that remained after 10 randomly selected trials in which the amount at risk ranged from CNY 15–35. Note that participants were informed that one Mu was equal to CNY 0.5 after the task had been completed. Participants were informed explicitly that Players A and B would not observe their punishment decisions. 

The entire experiment consisted of 120 trials divided into 4 blocks so that each condition contained exactly 80 trials. To ensure that participants fully understood the task, before the formal experiment all participants were given 12 practice trials with 2 trials in each condition. The experimental procedure is shown in Figure 1.

### 2.4. Statistical Analyses

The data were analyzed using SPSS 24.0. All descriptive statistics were expressed as “mean ± standard error.” A 2 (Time: time pressure, no time pressure) × 3 (Offer: Fair, Medium unfair, Unfair) repeated measures ANOVA was conducted with retention of money points for third-party punishment, average Mus for punishment, and RTs as dependent variables, respectively. Degrees of freedom for F-ratios were corrected using the Greenhouse–Geisser method when the assumption of sphericity was violated. Statistical differences were considered significant at *p* < 0.05, and post-hoc comparisons were Bonferroni-corrected at *p* < 0.05. In addition, a general linear mixed-effects model analysis was conducted on the participant retention data using the lme4 package [26,27] for the R programming environment, as participant choice (retention or penalty) was a binary variable.

## 3. Results

### 3.1. Retention Rate

The repeated measures ANOVA conducted on retention rate revealed a significant main effect of time, *F* (1,30) = 6.07, *p* = 0.02, $\eta_{p}^{2}\;$ = 0.17, such that retention rate was higher when time was the TP condition (TP: 58.12 ± 3.43%, NTP: 55.91 ± 3.33%, *p* = 0.02). There was a significant main effect of offer, *F* (1.62,48.56) = 88.12, *p* < 0.001, $\eta_{p}^{2}$ = 0.75, such that participants’ retention rate was higher when the offer was Fair (Fair: 93.63 ± 2.69%; Medium unfair: 60.81 ± 6.96%; High unfair: 16.61 ± 3.45%; Fair vs. Medium unfair, *p* < 0.001; Fair vs. High unfair, *p* < 0.001), and the Medium unfair offer retention rate was higher than Unfair (*p* < 0.001). Additionally, the interaction between time and offer was significant, *F* (1.56,46.89) = 6.49, *p* = 0.003, $\eta_{p}^{2}$ = 0.18. The results of the simple effects analysis showed that, for High unfair offers, the retention rate of TP was higher than NTP (TP: 21.15 ± 4.03%, NTP: 12.09 ± 3.31%, *p* = 0.002), and that when the offer was Fair or Medium unfair, there was no significant difference between TP and NTP (*p* > 0.5) (Figure 2).

The results of the general linear mixed-effects model analysis found that participants’ money retention choices increased under time pressure (TP: SE = 0.28, z = 2.38, *p* = 0.01). As the offers’ unfairness increased, participants’ money retention choices decreased (Medium unfair: SE = 0.26, z = 14.19, *p* < 0.001, Unfair: SE = 0.29, z = 24.02, *p* < 0.001). Crucially, when offers were relatively unfair and inequitable, time pressure for money retention options increased (Medium unfair: SE = 0.32, z = −2.21, *p* = 0.02, Unfair: SE = 0.33, z = 4.57, *p* < 0.001). These results were similar to those found in the ANOVA conducted on retention, with both suggesting that time pressure weakens participants’ punishment for violations.

### 3.2. Average MUs for Punishment

The repeated measure ANOVA conducted on average MUs for punishment revealed a significant main effect of offers, *F* (1.63,50.61) = 92.01, *p* < 0.001, $\eta_{p}^{2}$ = 0.75, such that more MUs were used for punishment when the offer was High unfair (Fair: 0.46 ± 0.1; Medium: 1.24 ± 0.13; High: 2.65 ± 0.15, Unfair vs. Fair, *p* < 0.001, Unfair vs. Medium unfair, *p* < 0.001). When the offer was Medium unfair, more Mus were used for punishment than for Fair offers (*p* < 0.001). No other main effect or interaction was significant (*p* > 0.05) (Figure 3).

### 3.3. Reaction Time

The repeated measures ANOVA conducted on RT revealed a significant main effect of time, *F* (1,30) = 15.78, *p* < 0.001, $\eta_{p}^{2}\;$ = 0.35, such that RT was faster when the time was the TP condition (TP:730.08 ± 20.67 ms, NTP: 924.86 ± 57.91 ms, *p* < 0.001). There was a significant main effect of offer, *F* (2,60) = 43.49, *p* < 0.001, $\eta_{p}^{2}$ = 0.59, such that participants’ RT was faster when the offer was Fair (Fair: 708.16 ± 29.29 ms, Medium unfair: 893.21 ± 45.81 ms, Unfair: 881.04 ± 37.46 ms, Fair vs. Medium, *p* < 0.001, Fair vs. High unfair, *p* < 0.001). Additionally, the interaction between time and offer was significant, *F* (2,60) = 6.98, *p* = 0.002, $\eta_{p}^{2}\;$ = 0.19, and further simple effects analysis revealed that TP was faster than NTP when the offer was under Fair (TP: 646.19 ± 21 ms; NTP: 770.12 ± 44.68 ms, *p* = 0.001), Medium unfair (TP: 772.06 ± 27.93 ms; NTP: 1014.36 ± 73.86 ms, *p* = 0.001), or High unfair (TP: 771.97 ± 22.56 ms; NTP: 990.11 ± 61.74 ms, *p* < 0.001) conditions (Figure 4).

To examine whether there were differences in response times between participants’ choices (keep or punish) under different conditions, we analyzed RT using participant choice as one of the variables. The repeated measures ANOVA on RT revealed a significant main effect of time, *F* (1,30) = 6.69, *p* = 0.015, $\eta_{p}^{2}\;$ = 0.18, such that RT was faster when time was the TP condition (TP:664.52 ± 36.01 ms, NTP: 760.32 ± 45.23 ms, *p* = 0.016). There was a significant main effect of offer, *F* (2,60) = 28.41, *p* < 0.001, $\eta_{p}^{2}$ = 0.49, such that participants’ RT was faster when the offer was Fair (Fair: 505.17 ± 38.51ms, Medium unfair: 850.66 ± 56.15 ms, High unfair: 781.42 ± 43.39 ms, Fair vs. Medium *p* < 0.001, Fair vs. High unfair *p* < 0.001). There was a significant main effect of choice, *F* (1,30) = 12.68, *p* < 0.001, $\eta_{p}^{2}$ = 0.29, such that RT was faster when the choice was punishment (punish: 649.05 ± 47.01ms; keep: 775.78 ± 33.99 ms, *p* = 0.002). Additionally, the interaction between offer and choice was significant, *F* (2,60) = 20.38, *p* < 0.001, $\eta_{p}^{2}\;$ = 0.41, and further simple effects analysis revealed that keep was faster than punishment when the offer was High unfair (keep: 697.89 ± 56.8 ms, punish: 877.75 ± 42.89 ms, *p* = 0.003). The interaction between time, offer, and choice was significant, *F* (1.62,45.46) = 5.64, *p* = 0.006, $\eta_{p}^{2}\;$ = 0.16, and further analysis revealed that TP was faster than NTP when the offer was Medium unfair when participants chose keep (TP: 773.72 ± 42.08 ms, NTP: 1031.52 ± 76.89 ms, *p* < 0.001). Additionally, TP was faster than NTP when the offer was High unfair when participants chose punish (TP: 762.66 ± 33.96 ms, NTP: 992.84 ± 64.02 ms, *p* < 0.001) (Figure 5).

## 4. Discussion

This study aimed to investigate whether individuals functioning as third parties would be more willing to maintain social norms under time pressure. Time pressure was imposed on the study participants by limiting the decision-making duration. The study’s results showed that participants’ responses were shorter for time pressure conditions compared with no time pressure conditions. With increasing levels of unfairness introduced, third-party money retention decreased in both the time pressure and no time pressure conditions as participants were shown to be willing to spend more MUs to punish the dictator for unfair offers as unfairness increased. While participants were willing to punish the dictator for unfair offers (offers that were both Medium unfair and High unfair), the retention of MUs in the High unfair condition was higher in the time pressure condition than in the no time pressure condition. Participants appeared to be less willing to punish the dictator under time pressure conditions compared with no time pressure conditions. Our results indicated that when there was time pressure to make a decision about High unfair proposals, individuals may have been driven to consider their own gains and losses rather than simply choosing to maintain social norms.

The study’s RTs results indicated that manipulation of time pressure and no time pressure conditions could induce differences in the performance of individuals’ responses. Responses were faster under conditions with time pressure compared with conditions without time pressure, further confirming participants’ verbal reports that selection pressure existed under conditions with time pressure. The study’s manipulation of time pressure was effective, as time pressure led participants to speed up their processing of information about the scenario and their selections. Participants accelerated their choices after perceiving the time pressure [28]. 

Player C’s retention of MUs decreased as the unfairness of the offer from Player A (dictator) to Player B increased. This result is in line with previous third-party decision-maker studies that indicated that participants were inclined to impose punishment when they observed an unfair offer [29,30]. As the unfairness of an offer increased, more punishment decisions were made. The third party was willing to take action to maintain social norms after injustice was observed. The results relating to the retention of MUs support the inequity aversion theory wherein inequity aversion drives people to punish fairness violations [29]. Moreover, the motivation to conform to social norms can lead to punishment internally (e.g., within one’s cultural group) if someone violates the norms of fairness [31]. The study also found higher MU retention rates when under time pressure compared with no time pressure. Especially in the High unfair condition, participants were shown to have significantly higher retention rates when they were under time pressure compared with when there was no time pressure. In contrast to previous studies [16,18,32], participants in this study did not show more pro-social behavior under time-pressure conditions when facing unfair offers, and retention of MUs was higher than under conditions without time pressure. Third parties under time pressure are likely to be more concerned about their own gains and losses than with the maintenance of social norms [15]. 

The dual-processing theory model suggests that individuals internalize social emotions and social norms in social interactions as an explicit dual process; specifically, individuals usually have competing choices when they make decisions, which tend to reflect one’s intuitive choice and one’s deliberate choice [33]. Utilitarian decision-making requires individuals to deliberate and overcome their moral intuition. Time restrictions placed on decision-making undoubtedly weaken people’s deliberative processes and drive them to make more selfish choices [12,14]. Time pressure interferes with the process of deliberation; thus, individuals behave more selfishly [19,34]. This result has been confirmed in recent research [35]. From a social reciprocity perspective, although imposing punishment for unfair offers builds one’s reputation and, thus, fair treatment by others in subsequent social interactions, concern about the threat of reprisals may lead individuals to avoid imposing punishment [36,37,38]. 

This study had several limitations. First, although participant RTs were measured and time pressure determined based on the manipulations in previous studies, there are differences in the perception of time pressure. Individuals, including study participants, may adapt to time pressure, which can eventually affect experimental results. Second, third-party altruistic behavior includes both punishment and assistance and choosing to punish or help are two distinct decisions. Thus, participants make different choices as a third party if they have more options (punish the dictator or compensate the victim) [35,39]. Finally, costly punishment is driven by two factors: the wrongdoer’s intentions and the harm caused to the victim [4,40,41,42]. Future research should further differentiate between the dictator’s intentions and the harm caused to the victim to explore the effects of time pressure on individuals’ maintenance of social norms under different intent–outcome combinations.

## 5. Conclusions

The study extended findings indicating that individuals under time pressure maintain social norms under third-party punishment conditions. The current study demonstrated that time pressure causes people to focus on self-interest in unjust situations and led participants to be more pro-self, which consequently weakened the behavior of maintaining social norms in highly unfair social situations. Thus, when faced with unfair situations, people may be able to overcome their selfish tendencies and ensure that social norms are maintained if they have sufficient time to do so.

## Figures and Tables

**Figure 1 brainsci-13-00227-f001:**
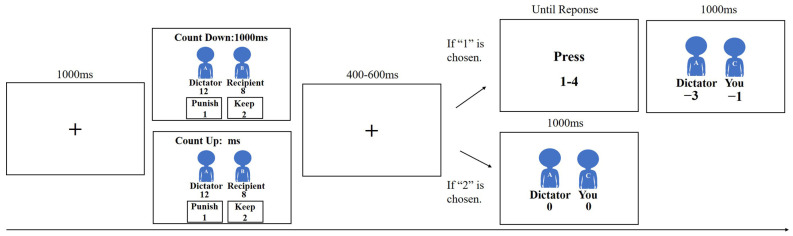
A depiction of the third-party punishment task.

**Figure 2 brainsci-13-00227-f002:**
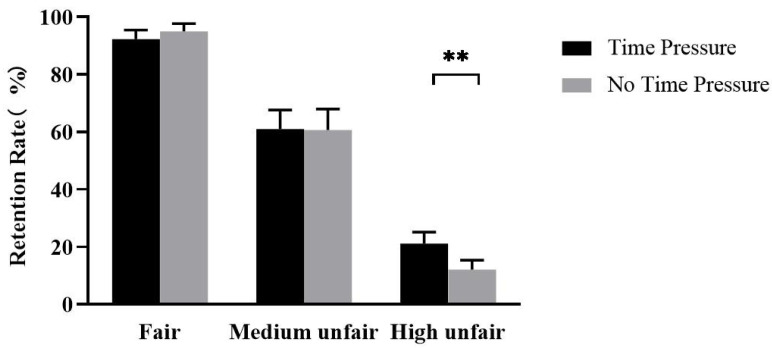
Histograms of the retention rates of different offers under time pressure and without time pressure (mean ± se). ** *p* < 0.01.

**Figure 3 brainsci-13-00227-f003:**
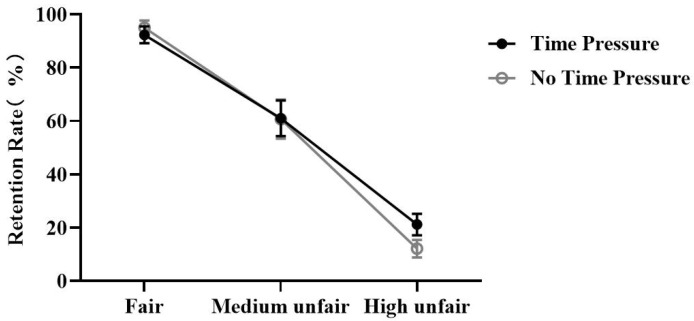
Line graph of retention rates for different offers under time pressure and no time pressure conditions (mean ± se).

**Figure 4 brainsci-13-00227-f004:**
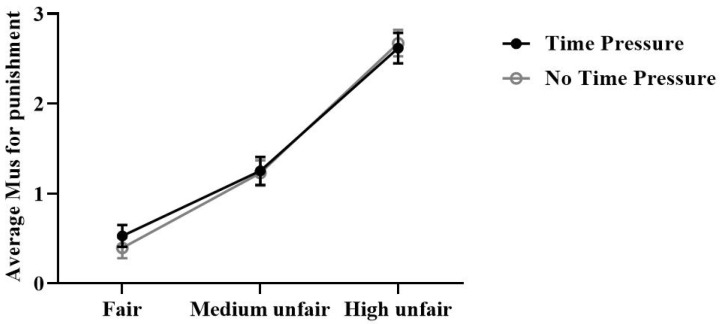
Participants’ average MUs spent to punish the dictator per round. (mean ± se).

**Figure 5 brainsci-13-00227-f005:**
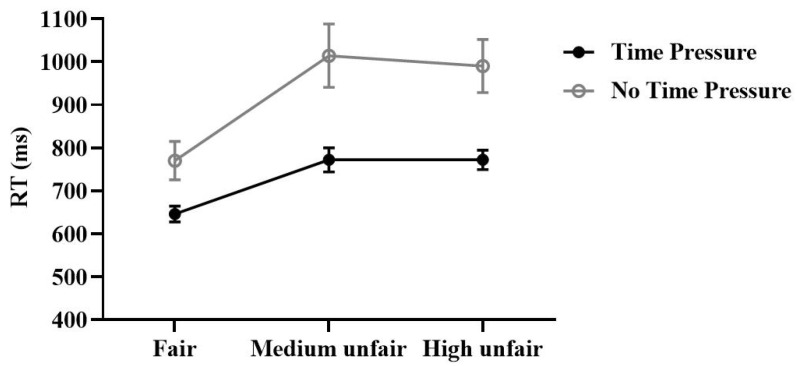
Response times of participants with different offers in time pressure and no time pressure conditions (mean ± se).

## Data Availability

The data are available from the corresponding author upon reasonable request.

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
