# Peer review of "Time Pressure Weakens Social Norm Maintenance in Third-Party Punishment"

_brainsci, 2023, doi:10.3390/brainsci13020227_

Round 1
Reviewer 1 Report
I don't understand the experimental design properly. Can the authors perhaps answer the following questions for me?
1) What choices do the dictators make exactly? Do they make binary choices? What is the strategy space?
2) Can the authors be a bit more precise about what the 2x3 treatments represent?
3) What do you mean with the ‘retention rate’?
4) Can I perhaps read the instructions that the subjects read?
Author Response
Dear Reviewer,
Thank you for your opinions, theses comments are very helpful to improve the quality of the manuscript. Now I response your comments with a point by point and highlight the changes in revised manuscript.
- What choices do the dictators make exactly? Do they make binary choices? What is the strategy space?
Response 1: In this experiment, the dictator was Player A and the recipient was Player B. Player A and Player B existed virtually, and in fact all subjects participated as Player C, a third party. The reason why each group of three completed the task was to increase the subjects' credibility of the experimental task (believing that the task was a three-person interaction task and each person played a different role) and to make the subjects believe that they were third party to the task and that the other two were participating as Player A and Player B respectively. Therefore, the dictator does not make any monetary distribution choices in this task.
- Can the authors be a bit more precise about what the 2x3 treatments represent?
Response 2: The present experiment used a 2 x 3 within-subjects experimental design. 2 indicates that the first variable has two levels and 3 indicates that the second variable has three levels. In the present study, time is the first independent variable with two levels: Time pressure, No time pressure; offer is the second independent variable with three levels: Fair, Medium unfair and High unfair. The different levels of these two independent variables were combined into six conditions, which were Time Pressure - Fair, Time Pressure - Medium Unfair, Time Pressure – High Unfair, No Time Pressure - Fair, No Time Pressure - Medium Unfair and No Time Pressure - High Unfair. For a description of the experimental design section, a specific description is given on the second page of the article, lines 100-105.
- What do you mean with the ‘retention rate’?
Response 3: In this experiment, the subject, as a third party, was presented with two options at the offer screen: Punish (button 1, i.e. choose to spend their current turn's money points to punish the dictator who made the money offer) Keep (button 2, i.e. choose to keep their current turn's money points). The subject was asked to make a choice between these two options. The retention rate refers to the percentage of all trials of the third-party punishment task in which the current subject ultimately chose to retain.
- Can I perhaps read the instructions that the subjects read?
Response 4: We are pleased to provide instructions for the subject's experiment.
In English: Welcome to participate in our experiment. Next, you (Player C) will complete a three-player interaction task with two other players (Player A and Player B) as a third party.
The task will proceed as follows: First, a "+" will be displayed in the centre of the screen. Player A will then divide the money between himself and Player B (The distribution scheme offered by Player A could be fair or unfair). You will see the allocation of money between Player A and Player B. You will then be required to make a decision to punish Player A for the proposed distribution or to keep the money you have for the current turn (you have 4 Mus per turn), press 1 if you choose to punish and 2 if you choose to keep. There are two cases of timing at this stage: countdown with a choice time constraint and count up without time constraint. If you chose Punish, you will then have to choose the specific amount of money you spend (1-4 buttons) to punish Player A. For every 1 Mu you spend to punish Player A, Player A will lose 3 times of money you spent (e.g., if you spend 1 Mu, Player A will lose 3 Mus; if you spend 2 Mus, Player A will lose 6 Mus; if you spend 3 Mus, Player A will lose 9 Mus; if you spend 4 Mus, Player A will lose 12 Mus). At the end the amount of money lost by player A and you (player C) for the current turn will be presented.
Attention: During this task, please make your own decisions based on what you actually think, and there are no right or wrong choices that you make. Yours payment was calculated as a constant basic payment (10 CNY) plus the money that remained after 10 randomly selected trials. If you understand the requirements listed above, please press Q to go to the next screen.
In Chinese:欢迎参加我们的实验。接下来,你将以第三方的身份(玩家C)和另外两名玩家(玩家A和玩家B)完成一个三人互动任务。
任务过程如下:首先,屏幕中央会呈现“+”;随后玩家A将一笔前在自己和玩家B之间分配(玩家A提供的分配方案有可能公平,也有可能不公平),你将会看到玩家A和玩家B金钱分配,此时需要你做出惩罚玩家A提出的分钱方案或保留你当前回合的金钱数额(每回合你拥有4元),如果选择惩罚请按1,如果选择保留请按2。在这一阶段可能会出现两种情况的计时:倒计时有选择时间限制,计时没有反应时间限制。如果你选择惩罚,接下来需要选择具体花费的金钱(1-4按键)惩罚玩家A,你每花费1元惩罚玩家A,玩家A会损失你花费点数的3倍(例如,你花费1元,玩家A会失去3元;你花费2元,玩家A会失去6元;你花费3元,玩家A会失去9元;你花费4元,玩家A会失去12元)。最后会呈现玩家A和你(玩家C)当前回合失去的金钱数额。
请注意:在该任务过程中,请根据自己的真实想法做出决定,你做出的选择没有对错之分。你的实验报酬包括两部分:基本报酬(10元)以及随机抽取的10个选择中你保留的金钱。如果明白以上要求,请按Q键进入下一界面。
We would like to thank the referee again for taking the time to review our manuscript.

Reviewer 2 Report
The authors investigated a decision-making pattern under time pressure in dictator-game with third-party punishment option. While overall impression from the manuscript is quite high, there are still a few minor suggestions needs to be implemented to improve it further.
Results, Pages 4-6. While the authors described in detail the statistical approach they've used (a repeated-measured ANOVA), there is no information on the multiple comparisons correction procedure required in this case. Please add the information whether the multiple comparisons correction procedure was used and, if not, please add the section to the manuscript to present your results after this correction was applied.
We hope to get more information on that which will positively contribute to the level of the manuscript.
Author Response
Dear Reviewer,
We would like to thank you for your careful reading, helpful comments, and constructive suggestions, which has significantly improved the presentation of our manuscript. Now I response your comments and highlight the changes in revised manuscript.
The authors investigated a decision-making pattern under time pressure in dictator-game with third-party punishment option. While overall impression from the manuscript is quite high, there are still a few minor suggestions needs to be implemented to improve it further.
Results, Pages 4-6. While the authors described in detail the statistical approach they've used (a repeated-measured ANOVA), there is no information on the multiple comparisons correction procedure required in this case. Please add the information whether the multiple comparisons correction procedure was used and, if not, please add the section to the manuscript to present your results after this correction was applied.
Response 1: Thank you very much for your comments. We have corrected the descriptions in the statistical analysis section (for corrections please see page 4 and lines 153-162 of the paper, in blue font) and added the results after correction for multiple comparisons based on your comments (pages 4-6 results, in blue font). Thank you again for your careful reading and comments, which are very important to improve the quality of our manuscript.
Modification section of Statistical analyses:
The data were analysed using SPSS 24.0. All descriptive statistics were expressed as "mean ± standard error". A 2 (time: time pressure, no time pressure) x 3 (offer: fair, medium unfair, unfair) repeated measures ANOVA was done with retention of money points for third party punishment, average Mus for punishment, and RTs as dependent variables, respectively. Degrees of freedom for F-ratios were corrected using the Greenhouse-Geisser method when the assumption of sphericity was violated. Statistical differences were considered significant at p < 0.05, and post-hoc comparisons were Bonferroni-corrected at p < 0.05. In addition, a general linear mixed-effects model analysis was done on the participant retention data using the lme4 package[24,25] for the R programming environment, as participant choice (retention or penalty) is a binary variable.
We would like to thank the referee again for taking the time to review our manuscript.

Round 2
Reviewer 1 Report
Summary
The authors conduct a lab experiment where third party players can choose to punish a dictator in a dictator game. The subjects either face a time constraint or not, or are shown fair/medium/unfair offers. The results show that less punishment is used under time pressure.
Main comments
Unfortunately, I would advice to reject this study. The sole reason is that deception is used. Subjects were told that there were two other players, Player A and Player B, who made decisions in a dictator game. In reality, however, there were no Players A and B, but only Players C. In the current write-up of the study, it is not clear to the reader that there were no Players A and B. This became apparent after communication with the authors. Deception in lab experiments is allowed in the economics discipline. Especially in this case, deception was not necessary. The experiment could have been performed with real Players A and B. Even if the authors insist on not using Players A and B, then still other methods could have ensured that there was no deception. For example, the authors could have used a strategy method, where subjects are asked to state their choice conditional on other, hypothetical, choices.
Author Response
Dear Reviewer,
Thanks to your comments for our manuscript.
Main comments
Unfortunately, I would advice to reject this study. The sole reason is that deception is used. Subjects were told that there were two other players, Player A and Player B, who made decisions in a dictator game. In reality, however, there were no Players A and B, but only Players C. In the current write-up of the study, it is not clear to the reader that there were no Players A and B. This became apparent after communication with the authors. Deception in lab experiments is allowed in the economics discipline. Especially in this case, deception was not necessary. The experiment could have been performed with real Players A and B. Even if the authors insist on not using Players A and B, then still other methods could have ensured that there was no deception. For example, the authors could have used a strategy method, where subjects are asked to state their choice conditional on other, hypothetical, choices.
Response:
Thanks to the reviewers for their comments.
We respectfully disagree with the reviewer's view that deception was used in this study. The reasons for this are as follows.
First, the three subjects in the experiment completed the task in different laboratories, even though Player A and Player B were not actually present and all subjects completed the task as a third party (Player C). The purpose of this was to make the subjects believe that they were indeed interacting with others, and also to make them believe that they were randomly assigned as Player C. This is also the classic research paradigm and methodology used in third-party punishment studies (relevant literature in blue), both to ensure that subjects believe that they are interacting with others and that their third-party identity is randomly assigned. In addition, at the end of the experiment we informed the subjects of the entire experimental setup and manipulation.
Secondly, in psychological experiments, to ensure the ecological validity of the experiment, the experimental situation is also fictitiously set up and the subjects are informed of the true purpose of the experiment and its intentions after the experiment was completed.
Finally, as the reviewers point out, economic decisions use deception, and this study required subjects to weigh the gains and losses of their own and others' interests, and this conflict of interest was real, which also involved a trade-off of personal economic interests, and thus our experimental manipulation was necessary. In summary, we do not agree with the reviewers that this study used deception.
Gerfo, E.L., Gallucci, A., Morese, R., Vergallito, A., Ottone, S., Ponzano, F., Locatelli, G., Bosco, F.M., & Lauro, L.J. (2019). The role of ventromedial prefrontal cortex and temporo-parietal junction in third-party punishment behavior. NeuroImage, 200, 501-510.
Cui, F., Wang, C., Cao, Q., & Jiao, C. (2019). Social hierarchies in third-party punishment: A behavioral and ERP study. Biological Psychology, 146.
Wang, L., Lu, X., Gu, R., Zhu, R., Xu, R., Broster, L.S., & Feng, C. (2017). Neural substrates of context‐ and person‐dependent altruistic punishment. Human Brain Mapping, 38.
Thank you very much.
your sincerely

Reviewer 2 Report
Thank you for implementing all necessary changes to improve the quality of the manuscript.
Author Response
Thank you again for your professional comments to improve the quality of our manuscript.